# The Impact of Photosynthetic Characteristics and Metabolomics on the Fatty Acid Biosynthesis in Tea Seeds

**DOI:** 10.3390/foods12203821

**Published:** 2023-10-18

**Authors:** Li Jiang, Shujing Liu, Xinrong Hu, Duojiao Li, Le Chen, Xiaoxing Weng, Zhaisheng Zheng, Xuan Chen, Jing Zhuang, Xinghui Li, Zhengdao Chen, Mingan Yuan

**Affiliations:** 1Jinhua Academy of Agricultural Sciences, Jinhua 321017, China; zhuwei2007100@163.com (L.J.); dajiangfang@163.com (X.H.); sweetwly2023@163.com (D.L.); chenle0103@163.com (L.C.); yuting11023@126.com (X.W.); zzs165@163.com (Z.Z.); 2International Institute of Tea Industry Innovation for the Belt and Road, Nanjing Agricultural University, Nanjing 210095, China; chenxuan@njau.edu.cn (X.C.); zhuangjing@njau.edu.cn (J.Z.); lxh@njau.edu.cn (X.L.); 3Zhejiang Cultivated Land Quality and Fertilizer Management Station, Hangzhou 310020, China; czd1989@126.com

**Keywords:** *Camellia sinensis*, fatty acid synthesis, Jincha 2 cultivar, Wuniuzao cultivar, targeted metabolomics, oil content, tea yield

## Abstract

The synthesis of tea fatty acids plays a crucial role in determining the oil content of tea seeds and selecting tea tree varieties suitable for harvesting both leaves and fruits. However, there is limited research on fatty acid synthesis in tea trees, and the precise mechanisms influencing tea seed oil content remain elusive. To reveal the fatty acid biosynthesis mechanism, we conducted a photosynthetic characteristic and targeted metabolomics analysis in comparison between Jincha 2 and Wuniuzao cultivars. Our findings revealed that Jincha 2 exhibited significantly higher net photosynthetic rates (Pn), stomatal conductance (Gs), and transpiration rate (Tr) compared with Wuniuzao, indicating the superior photosynthetic capabilities of Jincha 2. Totally, we identified 94 metabolites with significant changes, including key hormone regulators such as gibberellin A1 (GA1) and indole 3-acetic acid (IAA). Additionally, linolenic acid, methyl dihydrojasmonate, and methylthiobutyric acid, precursors required for fatty acid synthesis, were significantly more abundant in Jincha 2 compared with Wuniuzao. In summary, our research suggests that photosynthetic rates and metabolites contribute to the increased yield, fatty acid synthesis, and oil content observed in Jincha 2 when compared with Wuniuzao.

## 1. Introduction

*Camellia sinensis,* classified in the Camellia family and Camellia genus, is a perennial evergreen woody plant. Its leaves can be processed into tea, rich in beneficial compounds like tea polyphenols, amino acids, and caffeine, which exhibit anticancer, anti-inflammatory, and cardiovascular benefits [1]. The tea tree’s fruit, known as tea seeds, yields oil containing physiologically active components such as fatty acids, vitamin E, and squalene. This oil has shown promise for improving cognitive function and reducing neurotoxicity [2,3,4]. In an ovariectomized mouse model, tea seed oil, with its high levels of monounsaturated fatty acids, was found to prevent obesity and reduce fatigue compared with soybean oil and lard [5].

China’s tea gardens span over 47 million acres, yet the self-produced edible oil is less than 30%. In recent years, numerous studies have concentrated on enhancing tea seed yields, primarily by innovating tea tree cultivation models. However, there has been limited research on the development of tea seeds themselves. Fatty acid synthesis and regulation mechanisms have predominantly been explored in economic oil crops like rapeseed and soybeans [6,7,8], with woody oil plants focused on *Camellia oleifera* and walnuts [9]. A recent study revealed that the photosynthetic traits of plants significantly influence fruit quality, with metabolomics serving as a crucial tool in the study of these biological characteristics [10]. Utilizing mass spectrometry (MS) metabolomics analysis on tea at various developmental stages, it was found that changes in the carbon pool during leaf growth are related to fatty acid synthesis [11]. Studying the mechanism of fatty acid synthesis during the peak period of tea tree oil conversion holds significant potential for targeted cultivation and high yields in tea tree production.

Tea leaves are the primary source for producing tea beverages, while tea seeds are processed to extract tea oil. To optimize the utilization of tea seeds from the tea plant, we have chosen Wuniuzao and Jincha 2 cultivars for our study. Wuniuzao is the primary cultivar cultivated in Zhejiang province for both tea and tea seed oil production, whereas Jincha 2 is a novel cultivar bred by our institute that has better tea leaf quality, a higher yield of tea leaves, and a higher seed oil content compared with the Wuniuzao cultivar. Combining insights from photosynthetic characteristics and metabolomics research, we illuminated how the differences in photosynthetic efficiency and metabolism during the peak oil transformation period in Jincha 2 and Wuniuzao affect the biological characteristics, oil content, and fatty acid composition of mature tea seeds. We propose to lay a theoretical foundation for achieving a dual harvest of seeds and leaves, ultimately enhancing the overall benefits of tea gardens.

## 2. Materials and Methods

### 2.1. Experimental Conditions and Materials

The experimental site is located at the Jinhua Agricultural Science Research Institute, Shimen Farm, Wucheng District, Jinhua City, Zhejiang Province (north latitude 29.01452, east longitude 119.63130). This area has a subtropical monsoon climate, featuring warm and humid conditions and maintaining an average annual temperature of 17.0 °C. The terrain is level, characterized by red-yellow soil with a soil layer thickness exceeding 40 cm and a soil pH of 5.70. Both Jincha 2, a dual-harvest cultivar for both seeds and leaves, and Wuniuzao, the primary cultivated variety in Zhejiang, have 5 years of tea age. The tea trees were planted with a spacing of 1.5 m between plants and 2.0 m between rows.

### 2.2. Determination of Biological Characteristics, Oil Content, Individual Plant Yield, and Fatty Acid Composition of Mature Tea Fruits

We randomly selected 30 mature tea seeds with cracked outer skin from each of the one compartment, two compartments, and three compartment shapes. The transverse and longitudinal diameters of the seeds were measured using a 1/1000 vernier caliper. Then the seeds were dried at 100 °C in an electric blast drying oven for 6 h until there was no further change in the seed weight. Subsequently, we recorded the dry weight of the seeds using a 1/100 electronic balance. After removing the inner and outer skin of the tea seeds, the oil content was measured using a nuclear magnetic resonance instrument HCY-20 (Top Cloud-Agri, Hangzhou, China). We used refined tea seed oil with a moisture content of 0 as a reference standard.

For the fatty acid composition analysis, fatty acid methyl esters were prepared with the boron trifluoride methanol method [12], followed by separation using gas chromatography GC-4000 Plus (Tokyo, Japan) equipped with a flame ionization detector. In brief, we added 50 µL of internal standard (C17:0, 20 mg/mL) and 2.5 mL of a 20% boron trifluoride-methanol reagent (Macklin Chemical Reagent Co., Ltd., Shanghai, China) to pre-weighed samples (0.3 g dry weight) in a 15 mL centrifuge tube. The tube was sealed and heated at 70 °C for 30 min, followed by cooling and the addition of 1 mL of a 10% NaCl solution. Fatty acid methyl esters were subsequently extracted using 2 mL hexane (Macklin Chemical Reagent Co., Ltd., Shanghai, China) with the addition of 1 g Na_2_SO_4_ (Macklin Chemical Reagent Co., Ltd., Shanghai, China). Finally, the samples were centrifuged for 5 min at 2500 rpm, and the supernatant was collected for GC-MS analysis. The experimental conditions include using a CP-Sil 88 capillary column (0.25 mm × 60 mm, 0.25 μm, Agilent, Santa Clara, CA, USA) and nitrogen as the carrier gas. The column temperature was set at a rate of 4 °C/min from 140 °C to 180 °C and 2 °C/min from 180 °C to 225 °C. The flow rate was set at 1 mL/min, with an injection volume of 1 μL. The split ratio was 1:50, and both the injection and detector temperatures were maintained at 240 °C. The analysis involved identifying and comparing the retention times of fatty acids with those of standard fatty acid samples [13]. The standard comprises a mixture of 37 fatty acid methyl esters (http://www.anpel.com.cn/products_4083.html, accessed on 8 November 2020). Prior to measurement, we dilute isooctane to varying concentrations by a factor of 10.

### 2.3. Determination of Chlorophyll Content during the Peak Period of Tea Tree Oil Transformation

In early September, we selected 5 tea trees from each cultivar and collected 3 fully mature, tender leaves from each tea tree to measure chlorophyll content. The determination of chlorophyll content followed the method described by Lichtenthaler et al. [14]. Summarily, the leaves were punched with a 0.6 cm diameter punch and placed in an Eppendorf tube containing 20 mL of 95% ethanol extraction solution. The tubes were sealed and kept in darkness for 24 h. The colors at wavelengths of 665 nm, 649 nm, and 470 nm were compared using a spectrophotometer, and the concentrations of each component were calculated, including the chlorophyll a (Chl a), chlorophyll b (Chl b), total chlorophyll (Chl (a + b)), and the changes in chlorophyll a compared with chlorophyll b (Chl a/b). We repeated the experiment 3 times.

### 2.4. Determination of Photosynthesis during the Peak Period of Tea Tree Oil Transformation

We measured the photosynthetic quantum flux density (PPFD) curve of fully developed new leaves using a GFS-6400 portable photosynthetic analyzer (LI-COR Bioscience, Lincoln, NE, USA) on a clear and cloudless day at 10:00 am in early September. During the measurement process, the air temperature, CO_2_, and O_2_ concentrations were consistent with natural conditions. We measured 4 physiological indicators of tea trees, including net photosynthetic rate (Pn), stomatal conductance (Gs), intercellular CO2 concentration (Ci), and transpiration rate (Tr) [15]. We selected 5 tea trees from each variety and analyzed 15 leaves per plant.

### 2.5. Determination of Chlorophyll Fluorescence during the Peak Period of Tea Tree Oil Transformation

We measured the chlorophyll fluorescence using a portable chlorophyll fluorescence analyzer (WALZ, Effectrich, Germany). We measured the photochemical quantum yield (yield), electron transfer rate (ETR), photochemical quenching coefficient (qP), and fluorescence quenching parameters (qN and NPQ) of chlorophyll fluorescence within the range of 0–1200 umol·m^−2^·s^−1^ photosynthetic effective radiation (PAR) values. We used the Walz software (version v2.56zn, Walz, Efficient, Germany) [16] to analyze the data. These measurements were carried out on a clear and cloudless day at 10:00 am in early September.

### 2.6. Targeted Metabolomics Determination of Tea Fruit during the Peak Period of 1.5 Oil Conversion

Targeted metabolomics experiments were conducted as described in reference [17], with three biological replicates for each sample. Tea tree fruits were harvested in early September and subjected to freeze-drying. An amount of 20 mg freeze-dried tea fruit powder was mixed with 1 mL of extraction solution I (acetonitrile:isopropanol:water = 3:3:2). Additionally, we added 10 μL of 10 µM lidocaine (Macklin Chemical Reagent Co., Ltd., Shanghai, China) and 10-camphor sulfonic acid (Macklin Chemical Reagent Co., Ltd., Shanghai, China) as internal standards for positive and negative ion modes, respectively. After ultrasonic extraction to collect the supernatant, 1 mL of extraction solution II (acetonitrile:water = 1:1) was added to the remaining precipitate, followed by further ultrasonic extraction and collection of the supernatant. Then, 1 mL of extraction solution III (80% ethanol) was added to the precipitate, and another round of ultrasonic extraction was performed, with the supernatant collected. The supernatants were merged three times, freeze-dried, and reconstituted in 100 µL of 0.1% formic acid aqueous solution [18]. Targeted metabolomics based on High Performance Liquid Chromatography (HPLC)—Multiple Reaction Monitoring (MRM)—MS was carried out on an Agilent 1100 HPLC tandem AB Science 4000 QTRAP mass spectrometer using a C18 column (Agilent, Eclipse XDB-C18, 4.6 mm × 250 mm, 5 μm). The method for targeted metabolomics was developed by our collaborating laboratory, based on 338 standard samples, as detailed in reference [19]. We processed the raw mass spectrometry data and integrated peak area using MultiQuant software (version 2.1, AB SCIEX), normalizing with internal standard peaks. The identified metabolites were categorized, and pathway enrichment was performed using databases such as Kyoto Encyclopedia of Genes and Genomes (KEGG), ChemSpider, and PubChem.

### 2.7. Data Statistics and Analysis

The photosynthetic characteristics, biological characteristics of mature fruits, oil content, individual plant yield, and fatty acid composition were analyzed using SPSS statistical software (version 22.0; using IBM, Armonk, NY, USA). Targeted metabolomics data analysis using MetaboAnalyst 4.0 [20] included partial least squares discriminant analysis (PLS-DA) of metabolites, volcanic maps, and heat maps. Statistical significance was evaluated by the Student’s *t*-test when comparing two groups and the one-way analysis of variance (ANOVA) test when comparing multiple groups. All results were presented as mean±standard deviation (SD) from three independent biological replications. A *p*-value less than 0.05 was considered statistically significant.

## 3. Results

### 3.1. The Biological Characteristics of Matured Fruits in Jincha 2 and Wuniuzao

The fruits of Jincha 2 and Wuniuzao exhibit different shapes, ranging from spherical (one compartment) to renal (two compartments) and triangular (three compartments), as shown in Figure 1. In terms of composition within these compartments, Jincha 2 and Wuniuzao comprise 72.18% and 84.05% in one compartment, 18.79% and 11.44% in two compartments, and 9.03% and 4.51% in three compartments, respectively (Table 1). The one, two, and three compartment dimensions of Wuniuzao surpass those of Jincha 2 in both horizontal and vertical diameters (Table 1). Notably, the oil content, single plant seed yield, and single plant oil yield were significantly higher in Jincha 2 compared with Wuniuzao, although the fresh weight, dry weight, and moisture content of the one, two, and three compartments of Jincha 2 were lower than those of Wuniuzao (Table 2). This is possibly due to the fact that Jincha 2 possesses relatively thin fruit skin and plumper seeds, while Wuniuzao shows thicker fruit skin and comparatively thinner seeds.

### 3.2. Analysis of Fatty Acid Composition of Jincha 2 and Wuniuzao

Due to the difference in the seed oil content, we examined the total fatty acid content in fresh fruits and observed significant differences between Jincha 2 and Wuniuzao, measuring 154.52 g∙kg^−1^ and 65.13 g∙kg^−1^, respectively. Predominantly, unsaturated fatty acids constitute the major component. The fresh weights of monounsaturated fatty acids stand at 82.17 g∙kg^−1^ for Jincha 2 and 32.87 g∙kg^−1^ for Wuniuzao, primarily composed of oleic acid. Additionally, the fresh weights of polyunsaturated fatty acids are 43.75 g∙kg^−1^ in Jincha 2 and 21.17 g∙kg^−1^ in Wuniuzao, mainly composed of linoleic acid. Meanwhile, the fresh weights of saturated fatty acids are 28.60 g∙kg^−1^ and 11.09 g∙kg^−1^ in Jincha 2 and Wuniuzao, respectively, with the primary components being palmitic acid and stearic acid (Table 3). Notably, the ratio of monounsaturated to polyunsaturated fatty acids in tea seed oil aligns with the standards of modern nutritional and health oils. Additionally, it contains low levels of medium- and long-chain saturated fatty acids, which are beneficial for digestion and absorption.

### 3.3. Examination of Photosynthetic Characteristics of Jincha 2 and Wuniuzao during the Peak Period of Oil Transformation

Photosynthesis serves as the primary energy source for fruit development in plants. By measuring photosynthetic efficiency, we found that the net photosynthetic rate (Pn), stomatal conductance (Gs), and transpiration rate (Tr) of Jincha 2 exhibited higher levels compared with Wuniuzao (Table 4), with significant differences (*p* < 0.05). Furthermore, the content of chlorophyll a (Chl a) and chlorophyll b (Chl b) in Jincha 2 was higher than that of Wuniuzao (Table 5). Additionally, with the increase in photosynthetic effective radiation (PAR), Jincha 2 exhibited higher fluorescence yield (Yield), electron transfer rate (ETR), photochemical quenching (qP), and non-photochemical quenching (NPQ) compared with Wuniuzao (Figure 2), indicating that Jincha 2 has a higher electron transfer rate, thereby enhancing its net photosynthetic rate. This, in turn, facilitates the synthesis of more organic compounds and provides increased energy for material synthesis and transformation, resulting in the higher fatty acid and yield in Jincha 2.

### 3.4. Differences in Metabolites during the Peak Period of Oil Transformation between Jincha 2 and Wuniuzao

We conducted comparative metabolomics studies on Jincha 2 and Wuniuzao during the peak oil conversion period, employing principal component analysis (PCA) to assess the experimental results of both varieties. The PCA analysis showed that the replications from each group were well separated, and differences in metabolite levels were present between the two cultivars (Figure 3). In total, we identified and quantified 183 metabolites in Jincha 2 and Wuniuzao. Compared with Wuniuzao, Jincha 2 exhibited significant differences ninety-four metabolites (*p* < 0.05), with eighty-seven metabolite levels showing a decrease and seven showing an increase. These identified metabolites were categorized into 13 groups, including amino acids, nucleic acids, organic acids, and other compounds (Figure 4). Jincha 2 exhibited reduced levels of amino acids compared with Wuniuzao, and these amino acids mainly participate in the synthesis of secondary metabolites.

Furthermore, we conducted a Kyoto Encyclopedia of Genes and Genomes (KEGG) pathway enrichment analysis on the differential metabolites, examining the top 20 pathways with the most significant enrichment for examination. The results reveal significant enrichment in metabolic pathways such as aminoacyl-tRNA biosynthesis, glyoxylate and dicarboxylate metabolism, tricarboxylic acid (TCA) cycle, photosynthetic carbon fixation, zeatin biosynthesis, isoquinoline alkaloid synthesis, purine metabolism, glutathione metabolism, tyrosine metabolism, and pyrimidine metabolism (Figure 5). The involvement of cis-aconitic acid, citric acid, and malic acid in the TCA cycle, along with compounds like glutamine, uridine, uracil, and cytosine in the pyrimidine metabolism pathway, was significantly reduced in Jincha 2, may imply that acetyl-CoA and malonyl-CoA have a greater engagement in fatty acids synthesis (Figure 6). Additionally, compounds and precursors required for fatty acid synthesis, such as linolenic acid, methyl dihydrojasmonate, and methylthiobutyric acid, were significantly higher in Jincha 2 compared with Wuniuzao, indicative of the higher oil content in Jincha 2 (Figure 6).

### 3.5. Differential Analysis of Hormones during the Peak Period of Oil Conversion between Jincha 2 and Wuniuzao

From the metabolomics data, we found the content of gibberellin (GA-1), indoleacetic acid (IAA), and dihydrojasmonate methyl ester (DHJA-ME) were notably higher in Jincha 2 compared with Wuniuzao (Figure 7), with a significant difference (*p* < 0.05), suggesting GA-1, IAA, and DHJA-ME exhibited a promotive effect on increasing the oil content, individual plant yield, and oil production in Jincha 2. Conversely, the levels of abscisic acid glucose lipid (ABA-GE), abscisic acid (ABA), dihydrojasmonate (DHJA), jasmonate isoleucine (JA Ile), jasmonic acid (JA), and zeatin (ZR) were significantly lower in Jincha 2 compared with Wuniuzao (Figure 7), implying that ABA, JA, and ZR may exert a negative regulatory influence on the levels of oil content and production in tea plants. We conducted an analysis of the correlation between photosynthetic characteristics, hormone level, and oil content in tea seeds (Table 6) and summarized the positive regulators and possible negative regulators.

## 4. Discussion

During the peak oil transformation period, we found photosynthesis and metabolites as key factors contributing to the oil content in tea seeds. Leaves are the primary site for plant photosynthesis, serving as the basis for material accumulation. The accumulation and transport of materials are intricately linked to grain yield [21]. Jincha 2 exhibited a high net photosynthetic rate, actual photon yield, and electron transfer rate during the peak oil conversion phase, indicating its exceptional efficiency in harnessing light energy for the synthesis of photosynthetic products and fruit development.

The KEGG enrichment analysis unveiled significant differences in metabolites during the oil conversion period, particularly in the TCA cycle and pyrimidine metabolism (Figure 5). The TCA cycle, vital for energy generation, participates in numerous metabolic pathways such as hormone signaling and glycolysis [22,23]. Acetyl CoA can enter the TCA cycle to produce ATP [24] and serve as a precursor for fatty acid synthesis [25]. Interestingly, levels of aconitic acid, citric acid, and malic acid, key components of the TCA cycle, were significantly lower in Jincha 2 than those in Wuniuzao (Figure 6). However, Jincha 2 displayed a significantly higher net photosynthetic rate, actual photon yield, and electron transfer rate compared with Wuniuzao. This suggests that the photosynthetic metabolites in Jincha 2 more actively yield acetyl CoA, participating in fatty acid synthesis.

The biosynthesis of pyrimidine follows a highly conserved pathway across species, involving the conversion of orotic acid into uracil [26]. Furthermore, uracil oxidase catalyzes the transformation of uracil into malonic acid [27,28], which, in turn, synthesizes malonic acid coenzyme A with the assistance of methylmalonic acid coenzyme A synthase. Malonic acid coenzyme A functions as an essential metabolic intermediate of malonate [29] and contributes to fatty acid biosynthesis as a carbon donor substrate [30]. Our findings indicate a significant reduction in intermediate metabolites within the pyrimidine metabolism pathway in Jincha 2 compared with Wuniuzao (Figure 6). This suggests that in Jincha 2 fruit, malonic acid coenzyme A may be directed towards the synthesis of fatty acids in tea seeds.

Furthermore, fatty acid synthase (FAS) biosynthesizes fatty acids via Claisen-like condensations of malonyl-CoA, while polyketide synthases (PKS) share a similar biosynthetic pathway with FAS, utilizing the same precursors and cofactors. PKS also plays a pivotal role in the synthesis of secondary metabolites, such as flavonoid [31]. In our current study, we observed lower levels of secondary metabolites in the Jincha 2 cultivar compared with Wuniuzao, suggesting these synthases are more actively involved in fatty acid biosynthesis.

Plant hormones play an important role in the regulation of fruit development, orchestrating intricate interactions that govern various facets of this process [32]. GA-1 and IAA, classified as auxins, are known to stimulate plant growth and elevate crop yields [33]. Our study revealed higher levels of GA-1 and IAA in Jincha 2 compared with Wuniuzao. Previous studies have indicated that GA can promote leaf chlorophyll synthesis, prevent chlorophyll degradation, and increase chlorophyll content [34,35]. Chlorophyll, in turn, enhances crop photosynthesis, serving as the foundation of plant growth and development [36]. Furthermore, studies have illuminated the role of four plant hormones, including IAA and GA, in upregulating genes responsible for fatty acid synthesis in green algae, with IAA having the greatest stimulating effect on the fatty acid content in *Chlorella* [37].

Additionally, we observed reduced levels of ABA, ABA-GE, and zeatin in Jincha 2 compared with Wuniuzao. ABA, which is usually released from inactive ABA-glucose ester (ABA-GE), plays important roles in seed maturation processes [38], providing an explanation for the smaller fruit size observed in Jincha 2 compared with Wuniuzao. Zeatin, a prevalent cytokinin, typically stimulates plant growth by promoting cell division. Cytokinins act as antagonists to IAA in determining apical dominance [39]. Du et al. (2023) [40] indicated that fatty acid desaturase 2 (FAD2) inhibits cytokinin synthesis in peanuts using scRNA-seq, suggesting zeatin is negatively correlated with fatty acid biosynthesis. Collectively, Jincha 2 exhibited significantly higher levels of GA and IAA and lower levels of zeatin compared with Wuniuzao (Figure 7), potentially enhancing photosynthesis and resulting in the generation of more photosynthetic products in tea fruits. Consequently, this increases oil content and plant yields in Jincha 2 compared with Wuniuzao.

## 5. Conclusions

In summary, during the peak period of oil conversion, the higher net photosynthetic rate and chlorophyll fluorescence in Jincha 2 promote photosynthesis, facilitating the synthesis of more organic products. This, in turn, enhances fruit growth and quality, ultimately leading to a higher yield per plant at the maturity stage in Jincha 2 than in Wuniuzao. Through metabolomics analysis, we identified 94 metabolites with significant differences. Notably, Jincha 2 exhibited higher levels of GA1 and IAA, possibly contributing to increased levels of unsaturated fatty acids and higher oil content in Jincha 2 fruit by regulating photosynthesis. Additionally, the intermediate metabolites for fatty acid synthesis were increased in Jincha 2 compared with Wuniuzao. These findings provide a solid foundation for deeper exploration of fatty acid synthesis mechanisms among tea varieties, targeted improvement of tea breeding, and the realization of a double harvest of seeds and leaves.

## Figures and Tables

**Figure 1 foods-12-03821-f001:**
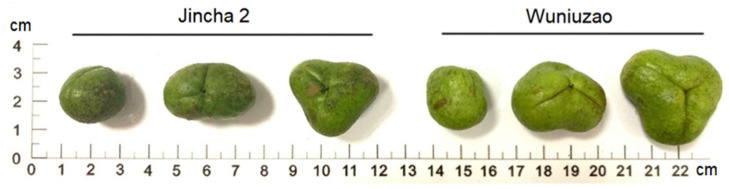
The ripe fruit of Jincha 2 and Wuniuzao cultivars. The ripe fruits are spherical (one compartment), renal (two compartments), and triangular (three compartments) in Jincha 2 and Wuniuzao.

**Figure 2 foods-12-03821-f002:**
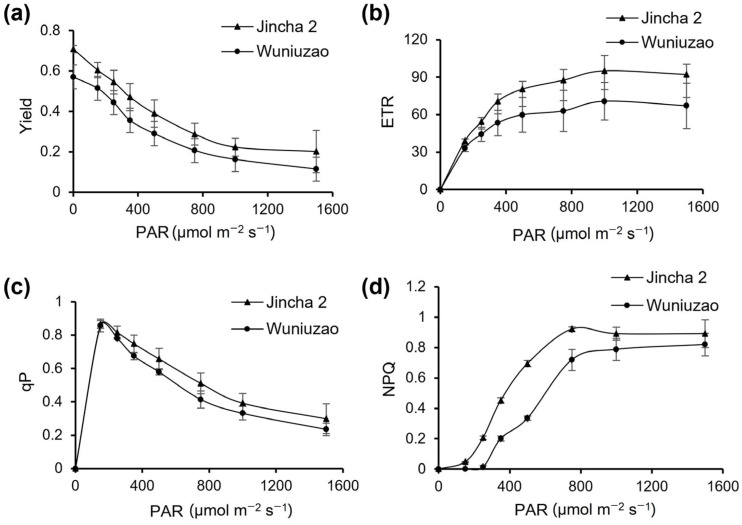
Photosynthesis efficiency analysis of Jincha 2 and Wuniuzao cultivars. (**a**) Changes in the effective quantum yield (Yield) of photochemical energy conversion in photosystem II in two cultivars. (**b**) Changes in the electron transport rate (ETR) of tea leaves in two cultivars. (**c**) Changes in the photochemical quenching (qP) value in two cultivars. (**d**) Changes in the non-photochemical quenching (NPQ) in two cultivars. PAR indicates the photosynthetic effective radiation.

**Figure 3 foods-12-03821-f003:**
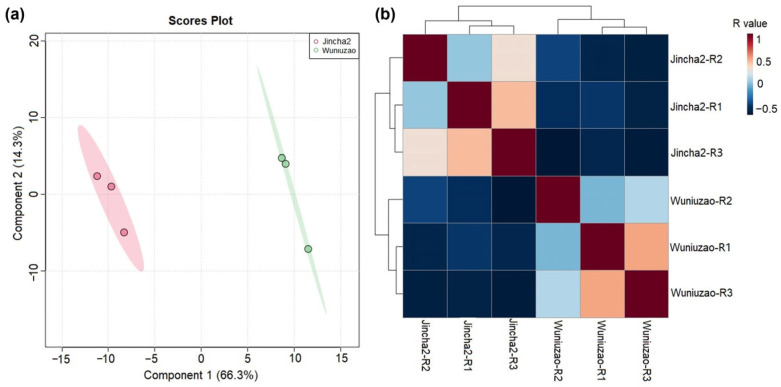
Principal component analysis (**a**) and correlation analysis (**b**) of the identified metabolites from Jincha 2 and Wuniuzao cultivars during the peak period of oil transformation. In the PCA analysis, the first principal component (PC1) represents the direction that accounts for the maximum variance in the data, while the second principal component (PC2) is orthogonal to PC1 and captures additional variance. R value means Pearson correlation coefficient.

**Figure 4 foods-12-03821-f004:**
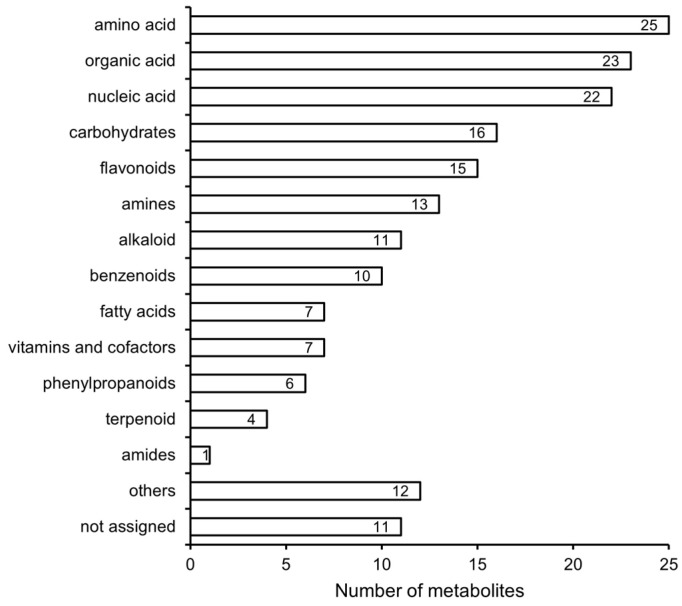
Classification of identified metabolites from Jincha 2 and Wuniuzao cultivars based on the KEGG database.

**Figure 5 foods-12-03821-f005:**
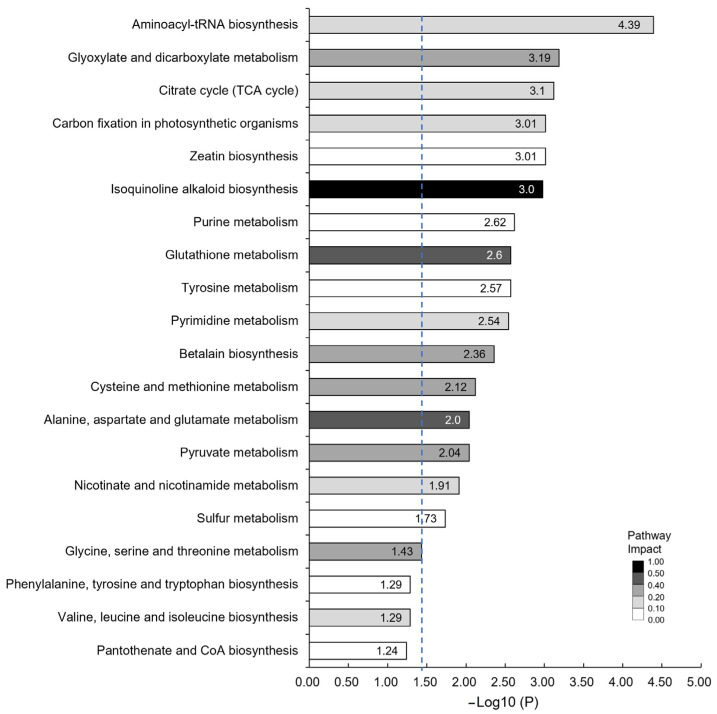
KEGG Pathway enrichment analysis of the significantly changed metabolites in tea fruits of Jincha 2 and Wuniuzao cultivars. Numbers inside the bars indicate the value of −Log_10_ (P). The scale bar of different levels of color indicate pathway impact. Blue dotted line represents *p* = 0.05.

**Figure 6 foods-12-03821-f006:**
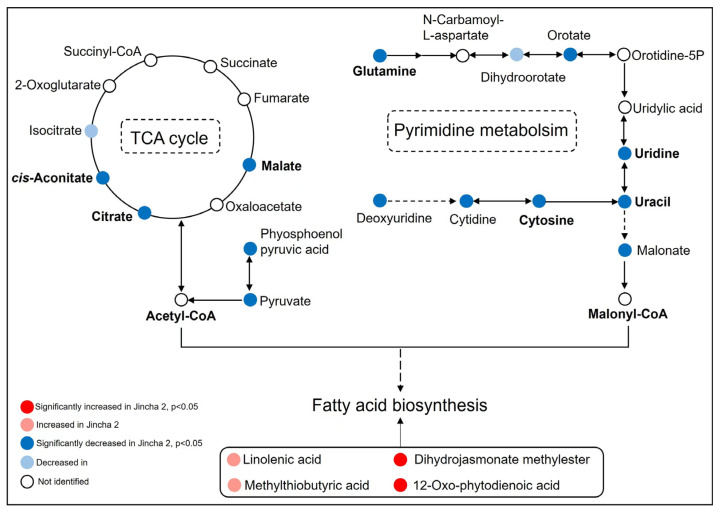
Differential metabolites in tricarboxylic acid cycle, pyrimidine metabolism, and fatty acid biosynthesis in Jincha 2 and Wuniuzao cultivars. The pathways are drawn based on the KEGG database. The detected metabolites are shown in a solid circle, and the undetected metabolites are shown in an empty circle. Colors indicate significance levels of the differential metabolites in Jincha 2 compared with Wuniuzao. Dashed lines indicate multiple steps.

**Figure 7 foods-12-03821-f007:**
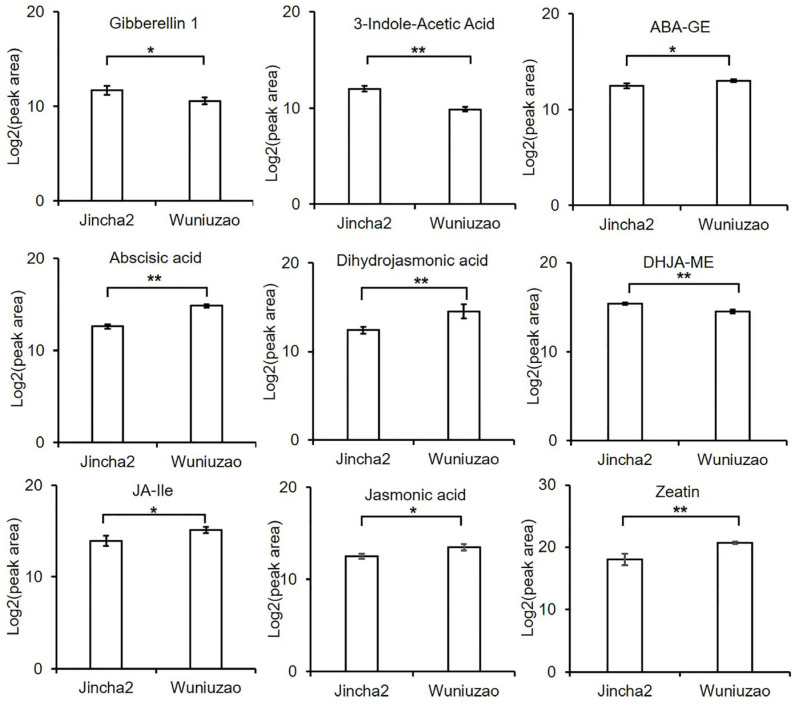
Levels of phytohormones in tea fruits of Jincha 2 and Wuniuzao cultivars during the peak period of oil transformation. Note: * indicates *p* < 0.05, ** indicates *p* < 0.01; JA-Ile, jasmonoyl-L-isoleucine; ABA-GE, abscisic acid glucose ester; DHJA: dihydrojasmonic acid; DHJA-ME: dihydrojasmonic acid methylester.

**Table 1 foods-12-03821-t001:** Biological characteristics of Jincha 2 and Wunizao cultivars.

Cultivars	One Compartment (Spherical)			Two Compartments (Renal)			Three Compartments (Triangular)		
	Percentage (%)	Height of Fruit (cm)	Diameter of Fruit (cm)	Percentage (%)	Height of Fruit (cm)	Diameter of Fruit (cm)	Percentage (%)	Height of Fruit (cm)	Diameter of Fruit (cm)
Jincha 2	72.18 ± 1.11%	1.81 ± 0.05	1.84 ± 0.08	18.79 ± 1.12% *	1.65 ± 0.17	2.79 ± 0.09	9.03 ± 0.2% *	1.76 ± 0.05	2.66 ± 0.07
Wuniuzao	84.05 ± 2.44% *	2.07 ± 0.06 *	2.22 ± 0.16 *	11.44 ± 0.9%	1.95 ± 0.16 *	3.20 ± 0.11 *	4.51 ± 1.5%	1.94 ± 0.1 *	2.89 ± 0.21 *

Note: Data are means ± standard error. The symbol of * indicates *p* < 0.05.

**Table 2 foods-12-03821-t002:** The productive yield of Jincha 2 and Wuniuzao cultivars.

Cultivars	One Compartment		Two Compartments		Three Compartments		Water Content (%)	Seed Oil Yield (%)	Single Plant Seed Yield (kg)	Single Plant Oil Yield (kg)
	Fresh Weight (g)	Dry Weight (g)	Fresh Weight (g)	Dry Weight (g)	Fresh Weight (g)	Dry Weight (g)				
Jincha 2	3.91 ± 0.03	1.23 ± 0.08	6.15 ± 0.01	2.62 ± 0.05	7.73 ± 0.07	3.44 ± 0.17	65.27%	35.18 ± 4.37% **	4.04 ± 0.27 **	1.42 ± 0.18 **
Wuniuzao	4.44 ± 0.12 *	1.34 ± 0.14 *	7.93 ± 0.11 **	2.83 ± 0.05 *	9.95 ± 0.11 *	4.20 ± 0.14 *	66.14% *	19.94 ± 1.78%	0.53 ± 0.06	0.11 ± 0.08

Note: Data are means ± standard error. The symbol of * indicates *p* < 0.05, ** indicates *p* < 0.01.

**Table 3 foods-12-03821-t003:** Fatty acid composition of tea seed oil in Jincha 2 and Wuniuzao.

No.	Compound Name	Formula	Contents in Jincha 2 (g∙kg^−1^ DW)	Percentage in Jincha 2 (%)	Contents in Wuniuzao (g∙kg^−1^ DW)	Percentage in Wuniuzao (%)
Unsaturated fatty acid						
1	Palmitoleic acid ^a^	C_16_H_30_O_2_	0.13 ± 0.004	0.08	0.14 ± 0.008	0.07
2	10-Heptadecenoic acid ^a^	C_17_H_32_O_2_	0.05 ± 0.001 *	0.03	0.03 ± 0.002	0.04
3	Oleic acid ^a^	C_18_H_34_O_2_	80.73 ± 1.61 *	52.32	32.48 ± 0.81	38.53
4	Linoleic acid ^b^	C_18_H_32_O_2_	43.2 ± 0.86 *	28	20.81 ± 0.5	24.69
5	Gadoleic acid ^a^	C_21_H_40_O_2_	1.17 ± 0.02 *	0.76	0.30 ± 0.01	0.36
6	α-Linolenic acid ^b^	C_18_H_30_O_2_	0.47 ± 0.01	0.3	0.53 ± 0.01 *	0.63
7	Eicosadienoic acid ^b^	C_20_H_36_O_2_	0.03 ± 0.002 *	0.02	0.02 ± 0.003	0.02
8	Arachidonic acid ^b^	C_20_H_32_O_2_	0.03 ± 0.004	0.02	0.03 ± 0.008	0.04
9	Nervonic acid ^b^	C_24_H_46_O_2_	0.02 ± 0.0001 *	0.01	n.d.	n.d.
Saturated fatty acid						
10	Myristic acid	C_14_H_28_O_2_	0.15 ± 0.002	0.1	0.27 ± 0.01 *	0.32
11	Pentadecylic acid	C_15_H_30_O_2_	0.022 ± 0.01	0.01	0.03 ± 0.005 *	0.04
12	Palmitic acid	C_16_H_32_O_2_	22.94 ± 0.41	14.87	26.44 ± 0.75 *	31.37
13	Erucic acid	C_22_H_42_O_2_	0.09 ± 0.002	0.06	0.08 ± 0.002	0.09
14	Lignoceric acid	C_24_H_48_O_2_	0.13 ± 0.003 *	0.08	0.11 ± 0.01	0.13
15	Stearic acid	C_18_H_36_O_2_	4.98 ± 0.1 *	3.23	2.97 ± 0.07	3.52
16	Arachidic acid	C_20_H_40_O_2_	0.15 ± 0.001 *	0.1	0.13 ± 0.04	0.15

Note: Data showed means ± standard deviation. The symbol of * indicates *p* < 0.05. ^a^ represents monounsaturated fatty acid; ^b^ represents polyunsaturated fatty acid; n.d., not detected.

**Table 4 foods-12-03821-t004:** Comparison of photosynthetic parameters of tea leaves between Jincha 2 and Wuniuzao.

Cultivars	Pn (umol∙m^−2^∙s^−1^)	Gs (mol∙m^−2^∙s^−1^)	Ci (umol∙m^−2^∙s^−1^)	Tr (m^−2^∙s^−1^)
Jincha 2	9.6 ± 0.3 *	0.37 ± 0.01 *	252.23 ± 1.48	12.39 ± 0.18 *
Wuniuzao	7.93 ± 0.43	0.3 ± 0	256.18 ± 2.4	11.15 ± 0.09

Note: Data showed means ± standard deviation. The symbol of * indicates *p* < 0.05.

**Table 5 foods-12-03821-t005:** Comparison of chlorophyll (Chl) content in the leaves of Jincha 2 and Wuniuzao.

Cultivars	Chl a (mg∙g^−1^)	Chl b (mg∙g^−1^)	Chl a + Chl b (mg∙g^−1^)	Chl a/Chl b
Jincha 2	3.79 ± 0.73	1.68 ± 0.12	5.47 ± 0.62	2.25 ± 0.57
Wuniuzao	3.09 ± 0.52	1.28 ± 0.19	4.38 ± 0.71	2.42 ± 0.06

Note: Data showed means ± standard deviation.

**Table 6 foods-12-03821-t006:** Correlation analysis of photosynthetic characteristics, hormone level, and oil content of tea plants during the peak period of oil conversion.

	Chl a	Chl b	Pn	Gs	Ci	Tr	Yield	ETR	GA-1	IAA	ABA-GE	ABA	DHJA-ME	JA-Ile	JA	ZR
Single plant Seed yield	0.607	0.821 *	0.954 **	0.992 **	−0.896 *	0.976 **	0.863 *	0.863 *	0.762	0.974 **	−0.809	−0.976 **	0.931 **	−0.818 *	−0.833 *	−0.974 **
Single plant oil yield	0.518	0.825 *	0.924 **	0.968 **	−0.897 *	0.937 **	0.911 *	0.911 *	0.789	0.931 **	−0.808	−0.978 **	0.907 *	−0.864 *	−0.817 *	−0.966 **

The symbol of * indicates *p* < 0.05, ** indicates *p* < 0.01.

## Data Availability

Data are contained within the article.

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
