# Peer review of "The Impact of Photosynthetic Characteristics and Metabolomics on the Fatty Acid Biosynthesis in Tea Seeds"

_foods, 2023, doi:10.3390/foods12203821_

Round 1

Reviewer 1 Report

For targeted metabolite analysis, did you add internal standard? If yes, what was it?

2.7. Statistical analysis: Please clearly state the number of replicates you used.

I have major concerns regarding the statistical analyses of data. Which approach did you use to compare the means? Parametric or non-parametric? 

Regarding your PCA, what are the two main factors which contributed to the separation (clustering)?

In the pathway (Fig. 6), where is the general phenylpropanoid pathway? 

During the peak oil transformation period, amino acids were found as the major differentially accumulated metabolites. What are the potential functions of these amino acids?

During the peak oil transformation period, what are the roles of different phytohormones which differ between Jincha 2 and Wuniuzao cultivars?

How can the findings from this study be applied to improve the quality of tea beverage?

Can ethylene play a role during the peak oil transformation period? Any information in the literature? 

It would be informative to the readers if the authors perform antioxidant activity assays, such as ABTS, FRAP, and DPPH. Also, total phenolic content (TPC) to be compared between the two cultivars.

Minor editing is suggested. 

Reviewer 2 Report

Manuscript is well prepared, but the content is poorly related to the scope of Foods journal (is more suitable for plant physiology journals).

 I propose some corrects:

1) please correct sentences in lines 81-83 and 108,

2) please add details in line 85,

3) please add reference in line 86,

4) Table 3 - Are fatty acids contents presented per oil or whole seeds? If per oil, what are the remaining mass ca 850 g/kg in Jincha 2 and ca 900 g/kg in Wuniuzao?

5) Table 3 -   the letters a and b should be written as superscripts

In my opinion English language is properly used.

Reviewer 3 Report

The manuscript "The impact of photosynthetic characteristics and metabolomics on the fatty acid biosynthesis in tea seeds" presents the biosynthesis of fatty acids in a high oil containing the cultivar Jincha 2.

Abstract: needs to be improved with clear information of the methodology used in the study;

Keywords: I suggest you insert new words, as some are already in the title of this manuscript;

Camellia sinensis : scientific name should be italicized;

In the introduction it is necessary to justify the use of the cultivars Jincha 2 and Wuniuzao;

Enter the geographic coordinates of the design;

Why in the identification of fatty acids was taken into account only the RI? What about mass spectra?

The comparison of the main secondary metabolites found in the samples was absent from the discussion;

The authors report that the experiment site presented acidic PH. This factor was not taken into account in the results and discussion of the study;

The conclusion was absent from the manuscript

Round 2

Reviewer 1 Report

Revision is accepted. 

Reviewer 3 Report

Improvements were made to the manuscript.